# Adversarial Training Methods for Semi-Supervised Text Classification

**Takeru Miyato[1,2]\*, Andrew M Dai[2], Ian Goodfellow[3]**
takeru.miyato@gmail.com, adai@google.com, ian@openai.com
[1] Preferred Networks, Inc., ATR Cognitive Mechanisms Laboratories, Kyoto University
[2] Google Brain
[3] OpenAI

## Abstract

Adversarial training provides a means of regularizing supervised learning algorithms while virtual adversarial training is able to extend supervised learning algorithms to the semi-supervised setting. However, both methods require making small perturbations to numerous entries of the input vector, which is inappropriate for sparse high-dimensional inputs such as one-hot word representations. We extend adversarial and virtual adversarial training to the text domain by applying perturbations to the word embeddings in a recurrent neural network rather than to the original input itself. The proposed method achieves state of the art results on multiple benchmark semi-supervised and purely supervised tasks. We provide visualizations and analysis showing that the learned word embeddings have improved in quality and that while training, the model is less prone to overfitting.

## 1 Introduction

*Adversarial examples* are examples that are created by making small perturbations to the input designed to significantly increase the loss incurred by a machine learning model (Szegedy et al., 2014; Goodfellow et al., 2015). Several models, including state of the art convolutional neural networks, lack the ability to classify adversarial examples correctly, sometimes even when the adversarial perturbation is constrained to be so small that a human observer cannot perceive it. *Adversarial training* is the process of training a model to correctly classify both unmodified examples and adversarial examples. It improves not only robustness to adversarial examples, but also generalization performance for original examples. Adversarial training requires the use of labels when training models that use a supervised cost, because the label appears in the cost function that the adversarial perturbation is designed to maximize. *Virtual adversarial training* (Miyato et al., 2016) extends the idea of adversarial training to the semi-supervised regime and unlabeled examples. This is done by regularizing the model so that given an example, the model will produce the same output distribution as it produces on an adversarial perturbation of that example. Virtual adversarial training achieves good generalization performance for both supervised and semi-supervised learning tasks.

Previous work has primarily applied adversarial and virtual adversarial training to image classification tasks. In this work, we extend these techniques to text classification tasks and sequence models. Adversarial perturbations typically consist of making small modifications to very many real-valued inputs. For text classification, the input is discrete, and usually represented as a series of high-dimensional one-hot vectors. Because the set of high-dimensional one-hot vectors does not admit infinitesimal perturbation, we define the perturbation on continuous word embeddings instead of discrete word inputs. Traditional adversarial and virtual adversarial training can be interpreted both as a regularization strategy (Szegedy et al., 2014; Goodfellow et al., 2015; Miyato et al., 2016) and as defense against an adversary who can supply malicious inputs (Szegedy et al., 2014; Goodfellow et al., 2015). Since the perturbed embedding does not map to any word and the adversary presumably does not have access to the word embedding layer, our proposed training strategy is no longer intended as a defense against an adversary. We thus propose this approach exclusively as a means of regularizing a text classifier by stabilizing the classification function.

---

\*This work was done when the author was at Google Brain.

We show that our approach with neural language model unsupervised pretraining as proposed by Dai & Le (2015) achieves state of the art performance for multiple semi-supervised text classification tasks, including sentiment classification and topic classification. We emphasize that optimization of only one additional hyperparameter $\epsilon$, the norm constraint limiting the size of the adversarial perturbations, achieved such state of the art performance. These results strongly encourage the use of our proposed method for other text classification tasks. We believe that text classification is an ideal setting for semi-supervised learning because there are abundant unlabeled corpora for semi-supervised learning algorithms to leverage. This work is the first work we know of to use adversarial and virtual adversarial training to improve a text or RNN model.

We also analyzed the trained models to qualitatively characterize the effect of adversarial and virtual adversarial training. We found that adversarial and virtual adversarial training improved word embeddings over the baseline methods.

## 2 MODEL

We denote a sequence of $T$ words as $\{w^{(t)}|t = 1, \ldots, T\}$, and a corresponding target as $y$. To transform a discrete word input to a continuous vector, we define the word embedding matrix $\boldsymbol{V} \in \mathbb{R}^{(K+1)\times D}$ where $K$ is the number of words in the vocabulary and each row $\boldsymbol{v}_k$ corresponds to the word embedding of the $i$-th word. Note that the $(K + 1)$-th word embedding is used as an embedding of an 'end of sequence (eos)' token, $\boldsymbol{v}_{\text{eos}}$. As a text classification model, we used a simple LSTM-based neural network model, shown in Figure 1a. At time step $t$, the input is the discrete word $w^{(t)}$, and the corresponding word embedding is $\boldsymbol{v}^{(t)}$. We additionally tried the bidirectional

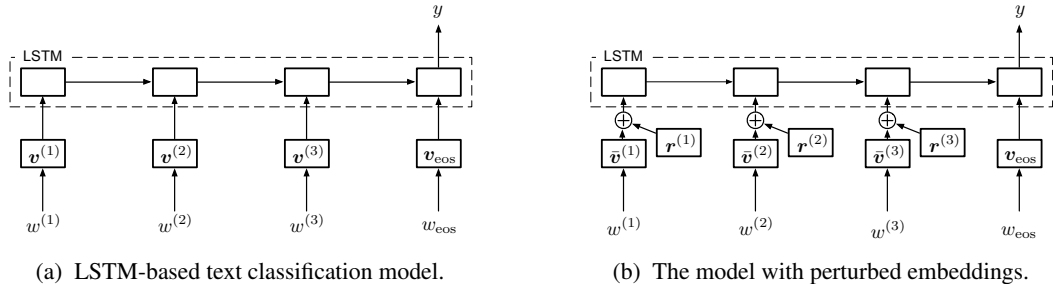

(a) LSTM-based text classification model.          (b) The model with perturbed embeddings.

Figure 1: Text classification models with clean embeddings (a) and with perturbed embeddings (b).

LSTM architecture (Graves & Schmidhuber, 2005) since this is used by the current state of the art method (Johnson & Zhang, 2016b). For constructing the bidirectional LSTM model for text classification, we add an additional LSTM on the reversed sequence to the unidirectional LSTM model described in Figure 1. The model then predicts the label on the concatenated LSTM outputs of both ends of the sequence.

In adversarial and virtual adversarial training, we train the classifier to be robust to perturbations of the embeddings, shown in Figure 1b. These perturbations are described in detail in Section 3. At present, it is sufficient to understand that the perturbations are of bounded norm. The model could trivially learn to make the perturbations insignificant by learning embeddings with very large norm. To prevent this pathological solution, when we apply adversarial and virtual adversarial training to the model we defined above, we replace the embeddings $\boldsymbol{v}_k$ with normalized embeddings $\bar{\boldsymbol{v}}_k$, defined as:

$$\bar{\boldsymbol{v}}_k = \frac{\boldsymbol{v}_k - \mathrm{E}(\boldsymbol{v})}{\sqrt{\mathrm{Var}(\boldsymbol{v})}} \text{ where } \mathrm{E}(\boldsymbol{v}) = \sum_{j=1}^{K} f_j \boldsymbol{v}_j, \mathrm{Var}(\boldsymbol{v}) = \sum_{j=1}^{K} f_j \left(\boldsymbol{v}_j - \mathrm{E}(\boldsymbol{v})\right)^2, \quad (1)$$

where $f_i$ is the frequency of the $i$-th word, calculated within all training examples.

## 3 ADVERSARIAL AND VIRTUAL ADVERSARIAL TRAINING

*Adversarial training* (Goodfellow et al., 2015) is a novel regularization method for classifiers to improve robustness to small, approximately worst case perturbations. Let us denote $\boldsymbol{x}$ as the input

and $\boldsymbol{\theta}$ as the parameters of a classifier. When applied to a classifier, adversarial training adds the following term to the cost function:

$$-\log p(y \mid \boldsymbol{x} + \boldsymbol{r}_{\text{adv}}; \boldsymbol{\theta}) \text{ where } \boldsymbol{r}_{\text{adv}} = \arg\min_{\boldsymbol{r}, \|\boldsymbol{r}\| \leq \epsilon} \log p(y \mid \boldsymbol{x} + \boldsymbol{r}; \hat{\boldsymbol{\theta}}) \qquad (2)$$

where $\boldsymbol{r}$ is a perturbation on the input and $\hat{\boldsymbol{\theta}}$ is a constant set to the current parameters of a classifier. The use of the constant copy $\hat{\boldsymbol{\theta}}$ rather than $\boldsymbol{\theta}$ indicates that the backpropagation algorithm should not be used to propagate gradients through the adversarial example construction process. At each step of training, we identify the worst case perturbations $\boldsymbol{r}_{\text{adv}}$ against the current model $p(y|\boldsymbol{x}; \hat{\boldsymbol{\theta}})$ in Eq. (2), and train the model to be robust to such perturbations through minimizing Eq. (2) with respect to $\theta$. However, we cannot calculate this value exactly in general, because exact minimization with respect to $\boldsymbol{r}$ is intractable for many interesting models such as neural networks. Goodfellow et al. (2015) proposed to approximate this value by linearizing $\log p(y \mid \boldsymbol{x}; \hat{\boldsymbol{\theta}})$ around $\boldsymbol{x}$. With a linear approximation and a $L_2$ norm constraint in Eq.(2), the resulting adversarial perturbation is

$$\boldsymbol{r}_{\text{adv}} = -\epsilon \boldsymbol{g}/\|\boldsymbol{g}\|_2 \text{ where } \boldsymbol{g} = \nabla_{\boldsymbol{x}} \log p(y \mid \boldsymbol{x}; \hat{\boldsymbol{\theta}}).$$

This perturbation can be easily computed using backpropagation in neural networks.

*Virtual adversarial training* (Miyato et al., 2016) is a regularization method closely related to adversarial training. The additional cost introduced by virtual adversarial training is the following:

$$\text{KL}[p(\cdot \mid \boldsymbol{x}; \hat{\boldsymbol{\theta}}) \| p(\cdot \mid \boldsymbol{x} + \boldsymbol{r}_{\text{v-adv}}; \boldsymbol{\theta})] \qquad (3)$$

$$\text{where } \boldsymbol{r}_{\text{v-adv}} = \arg\max_{\boldsymbol{r}, \|\boldsymbol{r}\| \leq \epsilon} \text{KL}[p(\cdot \mid \boldsymbol{x}; \hat{\boldsymbol{\theta}}) \| p(\cdot \mid \boldsymbol{x} + \boldsymbol{r}; \hat{\boldsymbol{\theta}})] \qquad (4)$$

where $\text{KL}[p\|q]$ denotes the KL divergence between distributions $p$ and $q$. By minimizing Eq.(3), a classifier is trained to be smooth. This can be considered as making the classifier resistant to perturbations in directions to which it is most sensitive on the current model $p(y|\boldsymbol{x}; \hat{\boldsymbol{\theta}})$. Virtual adversarial loss Eq.(3) requires only the input $\boldsymbol{x}$ and does not require the actual label $y$ while adversarial loss defined in Eq.(2) requires the label $y$. This makes it possible to apply virtual adversarial training to semi-supervised learning. Although we also in general cannot analytically calculate the virtual adversarial loss, Miyato et al. (2016) proposed to calculate the approximated Eq.(3) efficiently with backpropagation.

As described in Sec. 2, in our work, we apply the adversarial perturbation to word embeddings, rather than directly to the input. To define adversarial perturbation on the word embeddings, let us denote a concatenation of a sequence of (normalized) word embedding vectors $[\bar{\boldsymbol{v}}^{(1)}, \bar{\boldsymbol{v}}^{(2)}, \ldots, \bar{\boldsymbol{v}}^{(T)}]$ as $\boldsymbol{s}$, and the model conditional probability of $y$ given $\boldsymbol{s}$ as $p(y|\boldsymbol{s}; \boldsymbol{\theta})$ where $\boldsymbol{\theta}$ are model parameters. Then we define the adversarial perturbation $\boldsymbol{r}_{\text{adv}}$ on $\boldsymbol{s}$ as:

$$\boldsymbol{r}_{\text{adv}} = -\epsilon \boldsymbol{g}/\|\boldsymbol{g}\|_2 \text{ where } \boldsymbol{g} = \nabla_{\boldsymbol{s}} \log p(y \mid \boldsymbol{s}; \hat{\boldsymbol{\theta}}). \qquad (5)$$

To be robust to the adversarial perturbation defined in Eq.(5), we define the adversarial loss by

$$L_{\text{adv}}(\boldsymbol{\theta}) = -\frac{1}{N} \sum_{n=1}^{N} \log p(y_n \mid \boldsymbol{s}_n + \boldsymbol{r}_{\text{adv},n}; \boldsymbol{\theta}) \qquad (6)$$

where $N$ is the number of labeled examples. In our experiments, adversarial training refers to minimizing the negative log-likelihood plus $L_{\text{adv}}$ with stochastic gradient descent.

In virtual adversarial training on our text classification model, at each training step, we calculate the below approximated virtual adversarial perturbation:

$$\boldsymbol{r}_{\text{v-adv}} = \epsilon \boldsymbol{g}/\|\boldsymbol{g}\|_2 \text{ where } \boldsymbol{g} = \nabla_{\boldsymbol{s}+\boldsymbol{d}} \text{KL}\left[p(\cdot \mid \boldsymbol{s}; \hat{\boldsymbol{\theta}}) \| p(\cdot \mid \boldsymbol{s} + \boldsymbol{d}; \hat{\boldsymbol{\theta}})\right] \qquad (7)$$

where $\boldsymbol{d}$ is a $TD$-dimensional small random vector. This approximation corresponds to a 2nd-order Taylor expansion and a single iteration of the power method on Eq.(3) as in previous work (Miyato et al., 2016). Then the virtual adversarial loss is defined as:

$$L_{\text{v-adv}}(\boldsymbol{\theta}) = \frac{1}{N'} \sum_{n'=1}^{N'} \text{KL}\left[p(\cdot \mid \boldsymbol{s}_{n'}; \hat{\boldsymbol{\theta}}) \| p(\cdot \mid \boldsymbol{s}_{n'} + \boldsymbol{r}_{\text{v-adv},n'}; \boldsymbol{\theta})\right] \qquad (8)$$

where $N'$ is the number of both labeled and unlabeled examples.

See Warde-Farley & Goodfellow (2016) for a recent review of adversarial training methods.

## 4 EXPERIMENTAL SETTINGS

All experiments used TensorFlow (Abadi et al., 2016) on GPUs. Code will be available at https://github.com/tensorflow/models/tree/master/adversarial_text.

To compare our method with other text classification methods, we tested on 5 different text datasets. We summarize information about each dataset in Table 1.

IMDB (Maas et al., 2011)[1] is a standard benchmark movie review dataset for sentiment classification. Elec (Johnson & Zhang, 2015b)[2][3] is an Amazon electronic product review dataset. Rotten Tomatoes (Pang & Lee, 2005) consists of short snippets of movie reviews, for sentiment classification. The Rotten Tomatoes dataset does not come with separate test sets, thus we divided all examples randomly into 90% for the training set, and 10% for the test set. We repeated training and evaluation five times with different random seeds for the division. For the Rotten Tomatoes dataset, we also collected unlabeled examples using movie reviews from the Amazon Reviews dataset (McAuley & Leskovec, 2013) [4]. DBpedia (Lehmann et al., 2015; Zhang et al., 2015) is a dataset of Wikipedia pages for category classification. Because the DBpedia dataset has no additional unlabeled examples, the results on DBpedia are for the supervised learning task only. RCV1 (Lewis et al., 2004) consists of news articles from the Reuters Corpus. For the RCV1 dataset, we followed previous works (Johnson & Zhang, 2015b) and we conducted a single topic classification task on the second level topics. We used the same division into training, test and unlabeled sets as Johnson & Zhang (2015b). Regarding pre-processing, we treated any punctuation as spaces. We converted all words to lower-case on the Rotten Tomatoes, DBpedia, and RCV1 datasets. We removed words which appear in only one document on all datasets. On RCV1, we also removed words in the English stop-words list provided by Lewis et al. (2004)[5].

Table 1: Summary of datasets. Note that unlabeled examples for the Rotten Tomatoes dataset are not provided so we instead use the unlabeled Amazon reviews dataset.

|                 | Classes | Train   | Test   | Unlabeled | Avg. $T$ | Max $T$ |
|-----------------|---------|---------|--------|-----------|----------|---------|
| IMDB            | 2       | 25,000  | 25,000 | 50,000    | 239      | 2,506   |
| Elec            | 2       | 24,792  | 24,897 | 197,025   | 110      | 5,123   |
| Rotten Tomatoes | 2       | 9596    | 1066   | 7,911,684 | 20       | 54      |
| DBpedia         | 14      | 560,000 | 70,000 | –         | 49       | 953     |
| RCV1            | 55      | 15,564  | 49,838 | 668,640   | 153      | 9,852   |

### 4.1 RECURRENT LANGUAGE MODEL PRE-TRAINING

Following Dai & Le (2015), we initialized the word embedding matrix and LSTM weights with a pre-trained recurrent language model (Bengio et al., 2006; Mikolov et al., 2010) that was trained on both labeled and unlabeled examples. We used a unidirectional single-layer LSTM with 1024 hidden units. The word embedding dimension $D$ was 256 on IMDB and 512 on the other datasets. We used a sampled softmax loss with 1024 candidate samples for training. For the optimization, we used the Adam optimizer (Kingma & Ba, 2015), with batch size 256, an initial learning rate of 0.001, and a 0.9999 learning rate exponential decay factor at each training step. We trained for 100,000 steps. We applied gradient clipping with norm set to 1.0 on all the parameters except word embeddings. To reduce runtime on GPU, we used truncated backpropagation up to 400 words from each end of the sequence. For regularization of the recurrent language model, we applied dropout (Srivastava et al., 2014) on the word embedding layer with 0.5 dropout rate.

---

[1] http://ai.stanford.edu/~amaas/data/sentiment/

[2] http://riejohnson.com/cnn_data.html

[3] There are some duplicated reviews in the original Elec dataset, and we used the dataset with removal of the duplicated reviews, provided by Johnson & Zhang (2015b), thus there are slightly fewer examples shown in Table 1 than the ones in previous works(Johnson & Zhang, 2015b; 2016b).

[4] http://snap.stanford.edu/data/web-Amazon.html

[5] http://www.ai.mit.edu/projects/jmlr/papers/volume5/lewis04a/lyrl2004_rcv1v2_README.htm

For the bidirectional LSTM model, we used 512 hidden units LSTM for both the standard order and reversed order sequences, and we used 256 dimensional word embeddings which are shared with both of the LSTMs. The other hyperparameters are the same as for the unidirectional LSTM. We tested the bidirectional LSTM model on IMDB, Elec and RCV because there are relatively long sentences in the datasets.

Pretraining with a recurrent language model was very effective on classification performance on all the datasets we tested on and so our results in Section 5 are with this pretraining.

## 4.2 TRAINING CLASSIFICATION MODELS

After pre-training, we trained the text classification model shown in Figure 1a with adversarial and virtual adversarial training as described in Section 3. Between the softmax layer for the target $y$ and the final output of the LSTM, we added a hidden layer, which has dimension 30 on IMDB, Elec and Rotten Tomatoes, and 128 on DBpedia and RCV1. The activation function on the hidden layer was ReLU(Jarrett et al., 2009; Nair & Hinton, 2010; Glorot et al., 2011). For optimization, we again used the Adam optimizer, with 0.0005 initial learning rate 0.9998 exponential decay. Batch sizes are 64 on IMDB, Elec, RCV1, and 128 on DBpedia. For the Rotten Tomatoes dataset, for each step, we take a batch of size 64 for calculating the loss of the negative log-likelihood and adversarial training, and 512 for calculating the loss of virtual adversarial training. Also for Rotten Tomatoes, we used texts with lengths $T$ less than 25 in the unlabeled dataset. We iterated 10,000 training steps on all datasets except IMDB and DBpedia, for which we used 15,000 and 20,000 training steps respectively. We again applied gradient clipping with the norm as 1.0 on all the parameters except the word embedding. We also used truncated backpropagation up to 400 words, and also generated the adversarial and virtual adversarial perturbation up to 400 words from each end of the sequence.

We found the bidirectional LSTM to converge more slowly, so we iterated for 15,000 training steps when training the bidirectional LSTM classification model.

For each dataset, we divided the original training set into training set and validation set, and we roughly optimized some hyperparameters shared with all of the methods; (model architecture, batchsize, training steps) with the validation performance of the base model with embedding dropout. For each method, we optimized two scalar hyperparameters with the validation set. These were the dropout rate on the embeddings and the norm constraint $\epsilon$ of adversarial and virtual adversarial training. Note that for adversarial and virtual adversarial training, we generate the perturbation after applying embedding dropout, which we found performed the best. We did not do early stopping with these methods. The method with only pretraining and embedding dropout is used as the baseline (referred to as Baseline in each table).

## 5 RESULTS

### 5.1 TEST PERFORMANCE ON IMDB DATASET AND MODEL ANALYSIS

Figure 2 shows the learning curves on the IMDB test set with the baseline method (only embedding dropout and pretraining), adversarial training, and virtual adversarial training. We can see in Figure 2a that adversarial and virtual adversarial training achieved lower negative log likelihood than the baseline. Furthermore, virtual adversarial training, which can utilize unlabeled data, maintained this low negative log-likelihood while the other methods began to overfit later in training. Regarding adversarial and virtual adversarial loss in Figure 2b and 2c, we can see the same tendency as for negative log likelihood; virtual adversarial training was able to keep these values lower than other methods. Because adversarial training operates only on the labeled subset of the training data, it eventually overfits even the task of resisting adversarial perturbations.

Table 2 shows the test performance on IMDB with each training method. 'Adversarial + Virtual Adversarial' means the method with both adversarial and virtual adversarial loss with the shared norm constraint $\epsilon$. With only embedding dropout, our model achieved a 7.39% error rate. Adversarial and virtual adversarial training improved the performance relative to our baseline, and virtual adversarial training achieved performance on par with the state of the art, 5.91% error rate. This is despite the fact that the state of the art model requires training a bidirectional LSTM whereas our model only

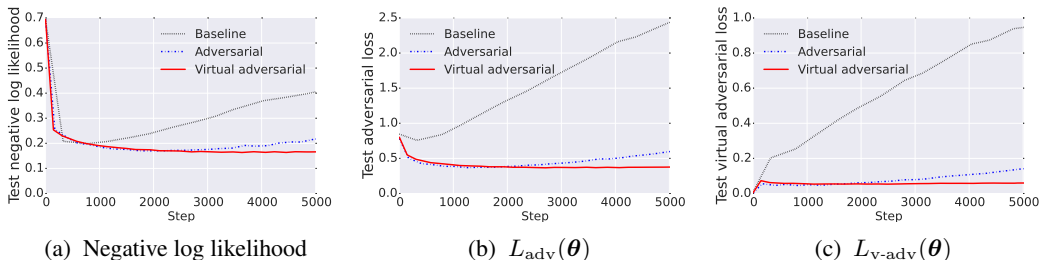

|  (a) Negative log likelihood | (b) $L_{\mathrm{adv}}(\boldsymbol{\theta})$ | (c) $L_{\mathrm{v\text{-}adv}}(\boldsymbol{\theta})$ |

Figure 2: Learning curves of (a) negative log likelihood, (b) adversarial loss (defined in Eq.(6)) and (c) virtual adversarial loss (defined in Eq.(8)) on IMDB. All values were evaluated on the test set. Adversarial and virtual adversarial loss were evaluated with $\epsilon = 5.0$. The optimal value of $\epsilon$ differs between adversarial training and virtual adversarial training, but the value of $5.0$ performs very well for both and provides a consistent point of comparison.

uses a unidirectional LSTM. We also show results with a bidirectional LSTM. Our bidirectional LSTM model has the same performance as a unidirectional LSTM with virtual adversarial training.

A common misconception is that adversarial training is equivalent to training on noisy examples. Noise is actually a far weaker regularizer than adversarial perturbations because, in high dimensional input spaces, an average noise vector is approximately orthogonal to the cost gradient. Adversarial perturbations are explicitly chosen to consistently increase the cost. To demonstrate the superiority of adversarial training over the addition of noise, we include control experiments which replaced adversarial perturbations with random perturbations from a multivariate Gaussian with scaled norm, on each embedding in the sequence. In Table 2, 'Random perturbation with labeled examples' is the method in which we replace $r_{\mathrm{adv}}$ with random perturbations, and 'Random perturbation with labeled and unlabeled examples' is the method in which we replace $r_{\mathrm{v\text{-}adv}}$ with random perturbations. Every adversarial training method outperformed every random perturbation method.

Table 2: Test performance on the IMDB sentiment classification task. * indicates using pretrained embeddings of CNN and bidirectional LSTM.

| Method | Test error rate |
|---|---|
| Baseline (without embedding normalization) | 7.33% |
| Baseline | 7.39% |
| Random perturbation with labeled examples | 7.20% |
| Random perturbation with labeled and unlabeled examples | 6.78% |
| Adversarial | 6.21% |
| Virtual Adversarial | **5.91**% |
| Adversarial + Virtual Adversarial | 6.09% |
| Virtual Adversarial (on bidirectional LSTM) | **5.91**% |
| Adversarial + Virtual Adversarial (on bidirectional LSTM) | 6.02% |
| Full+Unlabeled+BoW (Maas et al., 2011) | 11.11% |
| Transductive SVM (Johnson & Zhang, 2015b) | 9.99% |
| NBSVM-bigrams (Wang & Manning, 2012) | 8.78% |
| Paragraph Vectors (Le & Mikolov, 2014) | 7.42% |
| SA-LSTM (Dai & Le, 2015) | 7.24% |
| One-hot bi-LSTM* (Johnson & Zhang, 2016b) | 5.94% |

To visualize the effect of adversarial and virtual adversarial training on embeddings, we examined embeddings trained using each method. Table 3 shows the 10 top nearest neighbors to 'good' and 'bad' with trained embeddings. The baseline and random methods are both strongly influenced by the grammatical structure of language, due to the language model pretraining step, but are not strongly influenced by the semantics of the text classification task. For example, 'bad' appears in the list of

nearest neighbors to 'good' on the baseline and the random perturbation method. Both 'bad' and 'good' are adjectives that can modify the same set of nouns, so it is reasonable for a language model to assign them similar embeddings, but this clearly does not convey much information about the actual meaning of the words. Adversarial training ensures that the meaning of a sentence cannot be inverted via a small change, so these words with similar grammatical role but different meaning become separated. When using adversarial and virtual adversarial training, 'bad' no longer appears in the 10 top nearest neighbors to 'good'. 'bad' falls to the 19th nearest neighbor for adversarial training and 21st nearest neighbor for virtual adversarial training, with cosine distances of 0.463 and 0.464, respectively. For the baseline and random perturbation method, the cosine distances were 0.361 and 0.377, respectively. In the other direction, the nearest neighbors to 'bad' included 'good' as the 4th nearest neighbor for the baseline method and random perturbation method. For both adversarial methods, 'good' drops to the 36th nearest neighbor of 'bad'.

Table 3: 10 top nearest neighbors to 'good' and 'bad' with the word embeddings trained on each method. We used cosine distance for the metric. 'Baseline' means training with embedding dropout and 'Random' means training with random perturbation with labeled examples. 'Adversarial' and 'Virtual Adversarial' mean adversarial training and virtual adversarial training.

| | 'good' | | | | 'bad' | | | |
|---|---|---|---|---|---|---|---|---|
| | **Baseline** | **Random** | **Adversarial** | **Virtual Adversarial** | **Baseline** | **Random** | **Adversarial** | **Virtual Adversarial** |
| 1 | great | great | decent | decent | terrible | terrible | terrible | terrible |
| 2 | decent | decent | great | great | awful | awful | awful | awful |
| 3 | ×bad | excellent | nice | nice | horrible | horrible | horrible | horrible |
| 4 | excellent | nice | fine | fine | ×good | ×good | poor | poor |
| 5 | Good | Good | entertaining | entertaining | Bad | poor | BAD | BAD |
| 6 | fine | ×bad | interesting | interesting | BAD | BAD | stupid | stupid |
| 7 | nice | fine | Good | Good | poor | Bad | Bad | Bad |
| 8 | interesting | interesting | excellent | cool | stupid | stupid | laughable | laughable |
| 9 | solid | entertaining | solid | enjoyable | Horrible | Horrible | lame | lame |
| 10 | entertaining | solid | cool | excellent | horrendous | horrendous | Horrible | Horrible |

We also investigated the 15 nearest neighbors to 'great' and its cosine distances with the trained embeddings. We saw that cosine distance on adversarial and virtual adversarial training (0.159–0.331) were much smaller than ones on the baseline and random perturbation method (0.244–0.399). The much weaker positive word 'good' also moved from the 3rd nearest neighbor to the 15th after virtual adversarial training.

## 5.2 Test performance on Elec, RCV1 and Rotten Tomatoes dataset

Table 4 shows the test performance on the Elec and RCV1 datasets. We can see our proposed method improved test performance on the baseline method and achieved state of the art performance on both datasets, even though the state of the art method uses a combination of CNN and bidirectional LSTM models. Our unidirectional LSTM model improves on the state of the art method and our method with a bidirectional LSTM further improves results on RCV1. The reason why the bidirectional models have better performance on the RCV1 dataset would be that, on the RCV1 dataset, there are some very long sentences compared with the other datasets, and the bidirectional model could better handle such long sentences with the shorter dependencies from the reverse order sentences.

Table 5 shows test performance on the Rotten Tomatoes dataset. Adversarial training was able to improve over the baseline method, and with both adversarial and virtual adversarial cost, achieved almost the same performance as the current state of the art method. However the test performance of only virtual adversarial training was worse than the baseline. We speculate that this is because the Rotten Tomatoes dataset has very few labeled sentences and the labeled sentences are very short. In this case, the virtual adversarial loss on unlabeled examples overwhelmed the supervised loss, so the model prioritized being robust to perturbation rather than obtaining the correct answer.

## 5.3 Performance on the DBpedia purely supervised classification task

Table 6 shows the test performance of each method on DBpedia. The 'Random perturbation' is the same method as the 'Random perturbation with labeled examples' explained in Section 5.1. Note that

Table 4: Test performance on the Elec and RCV1 classification tasks. * indicates using pretrained embeddings of CNN, and $^{\dagger}$ indicates using pretrained embeddings of CNN and bidirectional LSTM.

| Method | Test error rate | |
|---|---|---|
| | Elec | RCV1 |
| Baseline | 6.24% | 7.40% |
| Adversarial | 5.61% | 7.12% |
| Virtual Adversarial | 5.54% | 7.05% |
| Adversarial + Virtual Adversarial | **5.40**% | 6.97% |
| Virtual Adversarial (on bidirectional LSTM) | 5.55% | 6.71% |
| Adversarial + Virtual Adversarial (on bidirectional LSTM) | 5.45% | **6.68**% |
| Transductive SVM (Johnson & Zhang, 2015b) | 16.41% | 10.77% |
| NBLM (Naıve Bayes logisitic regression model) (Johnson & Zhang, 2015a) | 8.11% | 13.97% |
| One-hot CNN* (Johnson & Zhang, 2015b) | 6.27% | 7.71% |
| One-hot CNN$^{\dagger}$ (Johnson & Zhang, 2016b) | 5.87% | 7.15% |
| One-hot bi-LSTM$^{\dagger}$ (Johnson & Zhang, 2016b) | 5.55% | 8.52% |

Table 5: Test performance on the Rotten Tomatoes sentiment classification task. * indicates using pretrained embeddings from word2vec Google News, and $^{\dagger}$ indicates using unlabeled data from Amazon reviews.

| Method | Test error rate |
|---|---|
| Baseline | 17.9% |
| Adversarial | 16.8% |
| Virtual Adversarial | 19.1% |
| Adversarial + Virtual Adversarial | **16.6**% |
| NBSVM-bigrams(Wang & Manning, 2012) | 20.6% |
| CNN*(Kim, 2014) | 18.5% |
| AdaSent*(Zhao et al., 2015) | 16.9% |
| SA-LSTM$^{\dagger}$ (Dai & Le, 2015) | 16.7% |

DBpedia has only labeled examples, as we explained in Section 4, so this task is purely supervised learning. We can see that the baseline method has already achieved nearly the current state of the art performance, and our proposed method improves from the baseline method.

Table 6: Test performance on the DBpedia topic classification task

| Method | Test error rate |
|---|---|
| Baseline (without embedding normalization) | 0.87% |
| Baseline | 0.90% |
| Random perturbation | 0.85% |
| Adversarial | 0.79% |
| Virtual Adversarial | **0.76**% |
| Bag-of-words(Zhang et al., 2015) | 3.57% |
| Large-CNN(character-level) (Zhang et al., 2015) | 1.73% |
| SA-LSTM(word-level)(Dai & Le, 2015) | 1.41% |
| N-grams TFIDF (Zhang et al., 2015) | 1.31% |
| SA-LSTM(character-level)(Dai & Le, 2015) | 1.19% |
| Word CNN (Johnson & Zhang, 2016a) | 0.84% |

# 6 RELATED WORKS

Dropout (Srivastava et al., 2014) is a regularization method widely used for many domains including text. There are some previous works adding random noise to the input and hidden layer during training, to prevent overfitting (e.g. (Sietsma & Dow, 1991; Poole et al., 2013)). However, in our experiments and in previous works (Miyato et al., 2016), training with adversarial and virtual adversarial perturbations outperformed the method with random perturbations.

For semi-supervised learning with neural networks, a common approach, especially in the image domain, is to train a generative model whose latent features may be used as features for classification (e.g. (Hinton et al., 2006; Maaløe et al., 2016)). These models now achieve state of the art performance on the image domain. However, these methods require numerous additional hyperparameters with generative models, and the conditions under which the generative model will provide good supervised learning performance are poorly understood. By comparison, adversarial and virtual adversarial training requires only one hyperparameter, and has a straightforward interpretation as robust optimization.

Adversarial and virtual adversarial training resemble some semi-supervised or transductive SVM approaches (Joachims, 1999; Chapelle & Zien, 2005; Collobert et al., 2006; Belkin et al., 2006) in that both families of methods push the decision boundary far from training examples (or in the case of transductive SVMs, test examples). However, adversarial training methods insist on margins on the input space , while SVMs insist on margins on the feature space defined by the kernel function. This property allows adversarial training methods to achieve the models with a more flexible function on the space where the margins are imposed. In our experiments (Table 2, 4) and Miyato et al. (2016), adversarial and virtual adversarial training achieve better performance than SVM based methods.

There has also been semi-supervised approaches applied to text classification with both CNNs and RNNs. These approaches utilize 'view-embeddings'(Johnson & Zhang, 2015b; 2016b) which use the window around a word to generate its embedding. When these are used as a pretrained model for the classification model, they are found to improve generalization performance. These methods and our method are complementary as we showed that our method improved from a recurrent pretrained language model.

# 7 CONCLUSION

In our experiments, we found that adversarial and virtual adversarial training have good regularization performance in sequence models on text classification tasks. On all datasets, our proposed method exceeded or was on par with the state of the art performance. We also found that adversarial and virtual adversarial training improved not only classification performance but also the quality of word embeddings. These results suggest that our proposed method is promising for other text domain tasks, such as machine translation(Sutskever et al., 2014), learning distributed representations of words or paragraphs(Mikolov et al., 2013; Le & Mikolov, 2014) and question answering tasks. Our approach could also be used for other general sequential tasks, such as for video or speech.

### ACKNOWLEDGMENTS

We thank the developers of Tensorflow. We thank the members of Google Brain team for their warm support and valuable comments. This work is partly supported by NEDO.

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
