# Peer review of "Adversarial Training Methods for Semi-Supervised Text Classification"

_ICLR 2017 — accepted_

[Official Review · AnonReviewer4 · rating 7 · confidence 5 · 15 Dec 2016]
**The paper reads well and has sufficent references. The application of adversarial training to text data is a simple but not trivial extension. The experimental section presents extensive experiments with comparison to alternative strategies. The proposed method is simple and effective and can be easily be applied after reading the paper.**

*** Paper Summary ***

This paper applies adversarial and virtual adversarial training to LSTM for text classification. Since text inputs are discrete adversarial perturbation are applied to the (normalized) word embeddings. Extensive experiments are reported and demonstrate the advantage of these methods.

*** Review Summary ***

The paper reads well and has sufficent references. The application of adversarial training to text data is a simple but not trivial extension. The experimental section presents extensive experiments with comparison to alternative strategies. The proposed method is simple and effective and can be easily be applied after reading the paper.

*** Detailed Review ***

The paper reads well. I have only a few comments regarding experiments and link to prior resarch:

Experiments:

- In Table 2 (and for other datasets as well), could you include an SVM baseline? e.g. S Wang and C Manning 2012?
- As another baseline, did you consider dropping words, i.e. masking noise? It is generally better than dropout/gaussian noise for text application (e.g. denoising autoencoders)?
- I am not sure I understand why virtual adversarial is worse than the baseline in Table 5. If you tune epsilon, in the worse case you would get the same performance as the baseline? Was it that validation was unreliable?

Related Work:

I think it would be interesting to point at SVM, transductive SVM who achieve something similar to adversarial training. When maximizing the margin in a (transductive) SVM, it is equivalent to move the example toward the decision boundary, i.e. moving them in the direction of increase of the loss gradient.

Also it would be interesting to draw a parallel between adversarial training and contrastive divergence. The adversarial samples are very close in nature to the one step Markov Chain samples from CD. See Bengio 2009. Related to this technique are also approaches that try to explicitely cancel the Jacobian at data points, e.g. Rifai et al 2011.

*** References ***

Marginalized Denoising Autoencoders for Domain Adaptation. Minmin Chen, K Weinberger.
Stacked Denoising Autoencoders. Pascal Vincent. JMLR 2011.
Learning invariant features through local space contraction, Salah Rifai, Xavier Muller, Xavier Glorot, Gregoire Mesnil, Yoshua Bengio and Pascal Vincent, 2011.
Learning Deep Architectures for AI, Yoshua Bengio 2009
Large Scale Transductive SVMs. Ronan Collobert et al 2006
Optimization for Transductive SVM.  O Chapelle, V Sindhwani, SS Keerthi JMLR 2008

[Official Review · AnonReviewer3 · rating 7 · confidence 3 · 17 Dec 2016]
**Good paper. The idea is simple but its result contributes new knowledge to the litterature**

This paper applies the idea of the adversarial training and virtual adversarial training to the LSTM-based model in the text context. The paper is in general well written and easy to follow. Extending the idea of the adversarial training to the text tasks is simple but non-trivial. Overall the paper is worth to publish. 

I only have a minor comment: it is also interesting to see how much adversarial training can help in the performance of RNN, which is a simpler model and may be easier to analyze.

[Official Review · AnonReviewer2 · rating 6 · confidence 4 · 18 Dec 2016]
**Reads well, missing theoretical differences with past techniques.**

The authors propose to apply virtual adversarial training to semi-supervised classification.

It is quite hard to assess the novelty on the algorithmic side at this stage: there is a huge available literature on semi-supervised learning (especially SVM-related literature, but some work were applied to neural networks too); unfortunately the authors do not mention it, nor relate their approach to it, and stick to the adversarial world.

In terms of novelty on the adversarial side, the authors propose to add perturbations at the level of words embeddings, rather than the input itself (having in mind applications to NLP).

Concerning the experimental section, authors focus on text classification methods. Again, comparison with the existing SVM-related literature is important to assess the viability of the proposed approach; for example (Wang et al, 2012) report 8.8% on IMBD with a very simple linear SVM (without transductive setup).

Overall, the paper reads well and propose a semi-supervised learning algorithm which is shown to work in practice. Theoretical and experimental comparison with past work is missing.

[Final Decision · Program Chairs · 06 Feb 2017]
**ICLR committee final decision**

This paper is concerned with extending adversarial and virtual adversarial training to text classification tasks. The main technical contribution is to apply perturbations to word embeddings rather than discrete input symbols. Excellent empirical performance is reported across a variety of tasks. 
 
 The reviewers were consensual in acknowledging the clarity and significance of the contribution, highlighting the quality of the numerical experiments. Moreover, the authors were responsive in the rebuttal phase and updated their paper with reviewers suggestions (such as the svm-related comparisons). 
 
 The AC thus recommends accepting this work as a poster.